# The Development of *Toxoplasma gondii* Recombinant Trivalent Chimeric Proteins as an Alternative to *Toxoplasma* Lysate Antigen (TLA) in Enzyme-Linked Immunosorbent Assay (ELISA) for the Detection of Immunoglobulin G (IgG) in Small Ruminants

**DOI:** 10.3390/ijms25084384

**Published:** 2024-04-16

**Authors:** Bartłomiej Tomasz Ferra, Maciej Chyb, Karolina Sołowińska, Lucyna Holec-Gąsior, Marta Skwarecka, Karolina Baranowicz, Justyna Gatkowska

**Affiliations:** 1Department of Tropical Parasitology, Institute of Maritime and Tropical Medicine in Gdynia, Medical University of Gdańsk, Powstania Styczniowego 9B, 81-519 Gdynia, Poland; karolina.baranowicz@gumed.edu.pl; 2Department of Molecular Microbiology, Faculty of Biology and Environmental Protection, University of Lodz, Banacha 12/16, 90-237 Lodz, Poland; maciej.chyb@edu.uni.lodz.pl (M.C.); justyna.gatkowska@biol.uni.lodz.pl (J.G.); 3Bio-Med-Chem Doctoral School of the University of Lodz and Lodz Institutes of the Polish Academy of Sciences, Faculty of Biology and Environmental Protection, University of Lodz, Banacha 12/16, 90-237 Lodz, Poland; 4Department of Molecular Biotechnology and Microbiology, Faculty of Chemistry, Gdańsk University of Technology, Narutowicza 11/12, 80-233 Gdansk, Poland; karolina.solowinska@pg.edu.pl (K.S.); lucyna.holec-gasior@pg.edu.pl (L.H.-G.); 5Institute of Biotechnology and Molecular Medicine, Kampinoska 25, 80-180 Gdansk, Poland; m.skwarecka@ibmm.pl

**Keywords:** *Toxoplasma gondii*, toxoplasmosis, recombinant chimeric protein, serodiagnosis, antibodies, ELISA, animal, sheep, goat

## Abstract

This study presents an evaluation of seventeen newly produced recombinant trivalent chimeric proteins (containing the same immunodominant fragment of SAG1 and SAG2 of *Toxoplasma gondii* antigens, and an additional immunodominant fragment of one of the parasite antigens, such as AMA1, GRA1, GRA2, GRA5, GRA6, GRA7, GRA9, LDH2, MAG1, MIC1, MIC3, P35, and ROP1) as a potential alternative to the whole-cell tachyzoite lysate (TLA) used in the detection of infection in small ruminants. These recombinant proteins, obtained by genetic engineering and molecular biology methods, were tested for their reactivity with specific anti-*Toxoplasma* IgG antibodies contained in serum samples of small ruminants (192 samples of sheep serum and 95 samples of goat serum) using an enzyme-linked immunosorbent assay (ELISA). The reactivity of six recombinant trivalent chimeric proteins (SAG1-SAG2-GRA5, SAG1-SAG2-GRA9, SAG1-SAG2-MIC1, SAG1-SAG2-MIC3, SAG1-SAG2-P35, and SAG1-SAG2-ROP1) with IgG antibodies generated during *T. gondii* invasion was comparable to the sensitivity of TLA-based IgG ELISA (100%). The obtained results show a strong correlation with the results obtained for TLA. This suggests that these protein preparations may be a potential alternative to TLA used in commercial tests and could be used to develop a cheaper test for the detection of parasite infection in small ruminants.

## 1. Introduction

*Toxoplasma gondii* is an intracellular, obligatory, protozoan parasite of all warm-blooded animals including humans and livestock. The parasite causes toxoplasmosis, a zoonosis with global distribution. The protozoan is considered one of the most important foodborne and waterborne parasites of veterinary importance, which is responsible for considerable economic losses annually. The protozoan causes asymptomatic infection in susceptible animals such as goats, sheep, pigs, or rabbits. Particularly in goats and sheep, the parasite invasion is associated with transplacental infection leading to mortality and morbidity in offspring, which cause significant reproductive losses. Furthermore, the infection of meat-producing farm animals poses a direct threat to human health, since the main route of human infection is the consumption of raw or underprepared meat containing tissue cysts of *T. gondii* [1].

The diagnosis of parasite infection relies primarily on serological evaluation of specific anti-*T. gondii* antibodies in serum. Usually, IgM and IgG antibodies are assessed using native *Toxoplasma* lysate antigen (TLA) which presents a few drawbacks. First of all, it is relatively expensive since a continuous culture of the parasite is needed to obtain it. Moreover, culture conditions and efficacy of parasite cell disruption greatly influence the actual antigenic composition of the lysate and the antigen itself consists of many native antigens, meaning that it is not fully characterized. Thus, each batch of the antigen requires standardization. Finally, the lysate fails to distinguish between the acute and chronic phases of toxoplasmosis, which is of paramount importance, especially for pregnant women. These problems can be solved using recombinant proteins that represent a relatively inexpensive, reliable, and fully defined diagnostic tool capable of distinguishing between invasion stages [2,3].

The widespread infectivity of *T. gondii* across various warm-blooded animals poses a significant epidemiological challenge. The only existing commercially available veterinary vaccine, ‘Toxovax’, has many limitations, including weakness and weight loss in sheep, as well as a lack of protection against horizontal parasite transmission, and is not intended for use on other species of parasite hosts [4,5].

Extensive research conducted over several decades has focused on unraveling the antigenic structure of *T. gondii*. This research has revealed which antigens play crucial roles in each stage of the parasite’s invasion into host cells. This knowledge has opened avenues for the development of drugs that can inhibit essential cellular mechanisms required during the initial invasion stages, as well as vaccines designed to induce the production of specific antibodies targeting parasite surface antigens or fragments that persist on infected host cells post-invasion. Antigens of *T. gondii* are distributed across various locations: the surface of the parasite’s cell membrane, the cytosol, secretory organelles (rhoptries, micronemes, dense granules), the parasitophorous vacuole, and tissue cysts. Research carried out so far has led to the discovery and identification of many parasite antigens [6,7,8,9,10,11,12]. Moreover, understanding the antigenic structure of the parasite has enabled the development of many recombinant proteins that can potentially be used as an alternative to TLA in the serodiagnosis of *T. gondii* invasion in humans [13] and animals [14].

The results of our previous studies, as well as those obtained by other research teams, prompted us to develop new diagnostic tools: recombinant chimeric proteins composed of various immunodominant fragments of different parasite antigens. Over many years, our research team has developed numerous single recombinant proteins with potential diagnostic significance. These recombinant proteins belong to four main families of parasite antigens, which are related in their ability to recognize, attach, invade, colonize, and multiply within host cells. The main families of parasite antigens include a) surface antigens (SAGs), e.g., SAG1, SAG2, SAG4, BSR4, and P35; b) rhoptry antigens (ROPs), e.g., ROP1; c) microneme antigens (MICs), e.g., MIC1, MIC3, and AMA1; d) high-density granule antigens (GRAs), e.g., GRA1, GRA2, GRA4, GRA5, GRA6, GRA7, and GRA9. Other antigens not assigned to any of the previous families are enzymes (for example, lactate dehydrogenase LDH1 and LDH2), bradyzoite cytoplasmic antigen (BAG1), or the major matrix antigen of tissue cysts (MAG1).

Our previous research on the use of recombinant proteins in the detection of parasite infections in farm animals has shown that this is a very complex problem [15,16,17]. A protein preparation well recognized by antibodies in one animal species may be completely useless for detecting infection in another. This is related to both the different immune responses in different animal species and the complex antigenic structure of the parasite itself. All the above encourages us to conduct further research on the development of a universal diagnostic algorithm for detecting parasite infection, for which recombinant chimeric proteins can be used.

The aim of this study was to construct and evaluate the diagnostic utility of seventeen newly produced recombinant trivalent chimeric proteins (all of them containing the same immunodominant fragment of SAG1 and SAG2 *T. gondii* antigens, and an additional immunodominant fragment of one of the parasite antigens, such as AMA1, GRA1, GRA2, GRA5, GRA6, GRA7, GRA9, LDH2, MAG1, MIC1, MIC3, P35, or ROP1) for the detection of *T. gondii* infection in small ruminants.

## 2. Results

### 2.1. Preliminary Evaluation of Single Recombinant Proteins Using IgG ELISA

Each of the twenty-four single recombinant proteins was coated in three concentrations (5, 2.5, and 1 µg/mL) and tested by IgG ELISA with three sera from naturally infected sheep and goats and with three samples from seronegative animals, respectively. The most representative results were obtained for plates coated with a 2.5 µg/mL dilution for individual protein preparations. In the case of both sheep and goat serum samples, at this protein concentration, the differences between positive and negative sera allowed for a clear distinction between them (Table 1, Appendix A). Two different dilutions of secondary antibodies were used. Regardless of the antibody dilution, the differences between positive and negative serum samples were very clearly visible. Expectedly, at a dilution of 1:16,000, the average values of the measured absorbance were higher than in the case of a dilution of 1:32,000. Not all protein preparations allowed for a clear differentiation between positive and negative serum samples. This was observed, among others, for AMA1C, BAG1, BSR4, GRA4, LDH1, MIC1ex3,4, and SAG4 preparations in the case of ovine sera. Specific anti-*T. gondii* antibodies were poorly detected in one or two of the positive sera and gave results similar to those obtained for the negative serum samples. For some protein preparations, the differences between positive and negative sera were low and were not multiples of the average values obtained for negative sera. In other cases, the results for all positive sera were high and differed significantly from the results obtained for negative sera, which gave similar results at a low level. For the remaining proteins, the average value obtained for the three positive sera was two or three times higher than the average value obtained for the negative sera. The same regularities were observed in the case of studies using goat sera. Once more, AMA1C, BAG1, BSR4, GRA4, LDH1, MIC1ex3,4, and SAG4 proteins were found to be the least effective in detecting specific anti-*T. gondii* antibodies. Due to the above results, AMA1, AMA1N, GRA1, GRA2ex2, GRA5, GRA6, GRA7, GRA9, LDH2, MAG1, MIC1, MIC1ex2, MIC3, P35, ROP1, SAG1, and SAG2 proteins were selected for the construction of new recombinant chimeric proteins. Based on the current state of knowledge and the results of our research, we decided to construct recombinant trivalent chimeric proteins containing the immunodominant fragments of the SAG1 and SAG2 *T. gondii* antigens and, as the third immunodominant fragment, one of the other parasite antigens selected during preliminary evaluation.

### 2.2. Plasmid Construction, Expression, and Purification of the Recombinant Chimeric Proteins

The nucleotide sequences of the resulting recombinant plasmids were confirmed by DNA sequencing (Genomed S.A., Warsaw, Poland). The characteristics of recombinant plasmids encoding recombinant chimeric proteins are shown in Appendix A. The size of the obtained plasmids ranged from 6657 to 8163 base pairs. The efficient expression occurs in both strains of *Escherichia coli* (Rosetta(DE3)pLysS and Rosetta(DE3)pLacI). However, higher expression levels of genes encoding recombinant chimeric proteins were observed in the Rosetta(DE3)pLysS strain of *E. coli*. An example of electrophoretic separation of proteins contained in lysates from bacterial cultures collected 16 h after expression induction is shown in Appendix A. The recombinant chimeric proteins were expressed as insoluble proteins with a calculated molecular mass between 50.15–106.08 kDa (Table 2) and were purified by one-step metal affinity chromatography. The full characteristics of recombinant chimeric proteins are shown in Table 2 and Appendix A. This expression system produces about 18–35 mg of purified proteins per liter of induced culture. Purification resulted in an electrophoretically homogeneous preparation with a purity above 90% (Appendix A).

### 2.3. Immunoreactivities of Mouse Anti-T. gondii IgG Antibodies in ELISA

As shown in Figure 1, primary invasion of the DX strain of *T. gondii* in BALB/c mice induced strong humoral IgG responses to a cocktail of native *T. gondii* antigens from the TLA preparation. In the case of the TLA preparation, the titer of specific IgG antibodies directed against individual parasite antigens is high from the early phase of invasion. For several recombinant chimeric proteins, such as SAG1-SAG2-AMA1, SAG1-SAG2-AMA1S, SAG1-SAG2-GRA1, SAG1-SAG2-GRA2, SAG1-SAG2-GRA9, SAG1-SAG2-LDH2, and SAG1-SAG2, a typical increase in the titer of IgG antibodies against particular antigens is observed as part of the development of a parasitic invasion. For these protein preparations, the standard immune response was observed. During *T. gondii* invasion, the reactivity of mouse sera was typically characterized by increasing IgG concentration until 6–12 weeks post-infection. Among these proteins, the worst response was observed in the case of the SAG1-SAG2-LDH2 and SAG1-SAG2-GRA1 protein preparations. In the case of recombinant chimeric proteins, SAG1-SAG2-GRA5, SAG1-SAG2-GRA6, SAG1-SAG2-GRA7, SAG1-SAG2-MAG1, SAG1-SAG2-MAG1S, SAG1-SAG2-MIC1, SAG1-SAG2-MIC1S, SAG1-SAG2-MIC3, SAG1-SAG2-P35, SAG1-SAG2-P35S, and SAG1-SAG2-ROP1 displayed strong humoral IgG responses in the early phase of *T. gondii* invasion and the responses remained at an almost constant level at all tested time points. The titer of IgG antibodies directed against the above-mentioned preparations was higher than for the SAG1-SAG2 preparation, except for the SAG1-SAG2-ROP1 protein. This suggests that the addition of another immunodominant fragment of *T. gondii* antigens increased the ability of these protein preparations to detect specific anti-*Toxoplasma* IgG antibodies. All recombinant chimeric proteins used displayed 100% IgG reactivity with mouse immune sera regardless of the time post-infection. The obtained results also show that the addition of another immunodominant fragment to the core of the recombinant chimeric protein structure may increase or decrease the reactivity of the protein preparation. This is very clear when comparing the results obtained for the SAG1-SAG2, SAG1-SAG2-LDH2, and SAG1-SAG2-GRA7 proteins.

The calculated cut-off values were: 0.3240 for SAG1-SAG2-AMA1, 0.3531 for SAG1-SAG2-AMA1S, 0.2016 for SAG1-SAG2-GRA1, 0.3644 for SAG1-SAG2-GRA2, 0.3346 for SAG1-SAG2-GRA5, 0.2877 for SAG1-SAG2-GRA6, 0.2769 for SAG1-SAG2-GRA7, 0.2551 SAG1-SAG2-GRA9, 0.3555 for SAG1-SAG2-LDH2, 0.2385 for SAG1-SAG2-MAG1, 0.2436 for SAG2-SAG2-MAG1S, 0.2630 for SAG1-SAG2-MIC1, 0.3145 for SAG1-SAG2-MIC1S, 0.2095 for SAG1-SAG2-MIC3, 0.3328 for SAG1-SAG2-P35, 0.3568 for SAG1-SAG2-P35S, 0.2414 for SAG1-SAG2-ROP1, 0.2377 for SAG1-SAG2, and 0.2027 for TLA.

### 2.4. Immunoreactivities of Ovine Anti-T. gondii IgG Antibodies in ELISA with Recombinant Chimeric Proteins and TLA

A total of 192 serum samples from sheep were examined. The seventeen recombinant trivalent chimeric proteins (all of them containing the same immunodominant fragment of SAG1 and SAG2 *T. gondii* antigens along with an additional immunodominant fragment of one of the parasite antigens, such as AMA1, GRA1, GRA2, GRA5, GRA6, GRA7, GRA9, LDH2, MAG1, MIC1, MIC3, P35, or ROP1), one recombinant divalent chimeric protein (SAG1-SAG2), and TLA reacted with different sensitivity, ranging between 94.55–100% (Table 3 and Figure 2). A 100% specificity calculated using ROC analysis for all tested seronegative serum samples was observed in the IgG ELISAs for all protein preparations. High reactivity, comparable to the sensitivity of TLA-based IgG ELISA (100%), was observed for the SAG1-SAG2, SAG1-SAG2-AMA1S, SAG1-SAG2-GRA1, SAG1-SAG2-GRA5, SAG1-SAG2-GRA9, SAG1-SAG2-MAG1, SAG1-SAG2-MIC1, SAG1-SAG2-MIC3, SAG1-SAG2-P35, and SAG1-SAG2-ROP1 protein preparations. Relatively high reactivity (98.18%) was noticed for three recombinant trivalent chimeric proteins, SAG1-SAG2-GRA2, SAG1-SAG2-LDH2, and SAG1-SAG2-MIC1S. Lower sensitivity was observed for the IgG ELISA based on the SAG1-SAG2-P35S (97.27%), SAG1-SAG2-GRA6 (96.36%), and SAG1-SAG2-MAG1S (96.36%) recombinant chimeric proteins. The lowest sensitivity, at a level of 94.55%, was observed for the IgG ELISA based on the SAG1-SAG2-AMA1 and SAG1-SAG2-GRA7 recombinant chimeric proteins.

### 2.5. Immunoreactivities of Caprine Anti-T. gondii IgG Antibodies in ELISA with Recombinant Chimeric Proteins and TLA

Taking into account the results previously obtained in IgG ELISA tests using mouse and sheep sera, further testing of protein preparations such as SAG1-SAG2-AMA1, SAG1-SAG2-AMA1S, SAG1-SAG2-LDH2, SAG1-SAG2-MAG1S, SAG1-SAG2-MIC1S, and SAG1-SAG2-P35S was abandoned. A total of 95 serum samples from goats were examined. The eleven recombinant trivalent chimeric proteins (all of them containing the same immunodominant fragment of SAG1 and SAG2 *T. gondii* antigens along with an additional immunodominant fragment of one of the parasite antigens, such as GRA1, GRA2, GRA5, GRA6, GRA7, GRA9, MAG1, MIC1, MIC3, P35, or ROP1), one recombinant divalent chimeric protein (SAG1-SAG2), and TLA reacted with different sensitivity, ranging between 90–100% (Table 4 and Figure 3). A 100% specificity calculated using ROC analysis for all tested seronegative serum samples was observed in the IgG ELISAs for all protein preparations. High reactivity, comparable to the sensitivity of TLA-based IgG ELISA (100%), was observed for the SAG1-SAG2-GRA2, SAG1-SAG2-GRA5, SAG1-SAG2-GRA9, SAG1-SAG2-MIC1, SAG1-SAG2-MIC3, SAG1-SAG2-P35, and SAG1-SAG2-ROP1 protein preparations. Relatively high reactivity (98%) was noticed for two recombinant chimeric proteins, SAG1-SAG2-GRA1 and SAG1-SAG2. Lower sensitivity, at a level of 96%, was observed for the IgG ELISA based on the SAG1-SAG2-GRA7 recombinant chimeric protein. The lowest sensitivity was observed for the IgG ELISA based on the SAG1-SAG2-MAG1 (92%) and SAG1-SAG2-GRA6 (90%) recombinant chimeric proteins.

### 2.6. Total Immunoreactivity of Small Ruminants’ Anti-T. gondii IgG Antibodies in ELISA with Recombinant Chimeric Proteins and TLA

Based on the results of IgG ELISA tests conducted on 287 serum samples from small ruminants (192 samples of sheep serum and 95 samples of goat serum), it was found that, out of the eleven recombinant trivalent chimeric proteins tested on both animal species, six (SAG1-SAG2-GRA5, SAG1-SAG2-GRA9, SAG1-SAG2-MIC1, SAG1-SAG2-MIC3, SAG1-SAG2-P35, and SAG1-SAG2-ROP1) exhibited high reactivity, comparable to the sensitivity of the TLA-based IgG ELISA (100%). Relatively high reactivity (99.38%) was noticed for two recombinant chimeric proteins, SAG1-SAG2-GRA1 and SAG1-SAG2, and at a level of 98.75% for SAG1-SAG2-GRA2. Lower sensitivity, at 97.50%, was observed for the IgG ELISA based on the SAG1-SAG2-MAG1 recombinant trivalent chimeric protein. The lowest sensitivity was observed for the IgG ELISA based on the SAG1-SAG2-GRA7 (95%) and SAG1-SAG2-GRA6 (94.38%) protein preparations. The Pearson product-moment correlation coefficient analysis (Table 5) showed a strong (0.70–0.89) or very strong (0.90–1.00) correlation between results obtained for recombinant proteins vs. TLA results. Furthermore, correlation analysis between recombinant trivalent chimeric proteins and the recombinant divalent SAG1-SAG2 protein showed a strong or very strong correlation between the results. The exception to this rule was observed with the protein preparations SAG1-SAG2-GRA5, SAG1-SAG2-MAG1, and SAG1-SAG2-ROP1, which were characterized by moderate correlation (0.50–0.69) in the studies using sheep serum samples.

## 3. Discussion

The prevalence of toxoplasmosis among animals is a significant diagnostic problem, especially from the point of view of epidemiological studies conducted in recent years, which concerned the detection of parasite infection in an increasing number of animal species, including aquatic mammals. On the one hand, the validity of epidemiological studies in animals is often due to purely economic reasons, a perfect example of which is sheep, in which parasite infection is associated with serious reproductive losses. On the other hand, monitoring the epidemiological situation is imposed by appropriate legal regulations, such as European Union Directive No. 2003/99/EC, which obliges the member countries to introduce programs enabling the determination of the incidence of parasitic diseases, especially among farm animals [18,19,20]. Unfortunately, in most countries, this is not followed by specific actions that would enable the determination of the real impact of the problem. The available data often do not reflect the actual epidemiological situation of specific parasitic diseases in the entire country but concern specific regions in which research was carried out by a scientific center (e.g., an institute or university). From the point of view of public health, diagnostics of toxoplasmosis in farm animals seem to be crucial in eliminating the main source of transmission of the parasite to humans, which is the consumption of meat from infected animals.

Currently, the diagnosis of toxoplasmosis in animals primarily relies on various types of agglutination tests (e.g., LAT and MAT) based on native antigens of the parasite which are expensive and therefore not suitable for testing large animal populations. Interestingly, indirect serological tests such as ELISA have not been widely used in the diagnosis of parasite infection in animals. This is probably due to the need for specialized equipment such as a microplate washer and a microplate reader, as well as the use of species-specific secondary antibodies. As mentioned earlier, the production of native antigens is expensive, time-consuming, and labor-intensive, requiring standardization of tests based on new batches of antigens. The solution to these challenges is the production of recombinant proteins in well-characterized expression systems, which is fast, cheap, and ensures consistent, defined protein preparations.

The aim of this study was to construct and evaluate the diagnostic utility of seventeen newly produced recombinant trivalent chimeric proteins (all of them containing the same immunodominant fragment of *T. gondii* SAG1 and SAG2 antigens, along with an additional immunodominant fragment of one of the parasite antigens, such as AMA1, GRA1, GRA2, GRA5, GRA6, GRA7, GRA9, LDH2, MAG1, MIC1, MIC3, P35, and ROP1) for the detection of *T. gondii* infection in small ruminants. The immunodominant fragments of parasite antigens were selected based on preliminary studies using single recombinant proteins in the IgG ELISA test. The selection of antigens to be included in the recombinant chimeric proteins was based on their ability to interact with specific anti-*Toxoplasma* antibodies in ovine and caprine serum samples. [9,21,22,23,24,25].

Over the last 30 years, the diagnostic utility of many recombinant *T. gondii* proteins has been assessed for the detection of specific antibodies in animal sera samples [14]. The results of some research teams support the potential use of recombinant proteins to detect specific anti-*Toxoplasma* antibodies in animal sera. In this study, we developed IgG ELISAs based on recombinant trivalent chimeric proteins, which demonstrated very high sensitivity and specificity with serum samples from small ruminants. Among the new recombinant proteins that we produced, high reactivity, comparable to the sensitivity of TLA-based IgG ELISA (100%), was observed for six recombinant trivalent chimeric proteins (SAG1-SAG2-GRA5, SAG1-SAG2-GRA9, SAG1-SAG2-MIC1, SAG1-SAG2-MIC3, SAG1-SAG2-P35, and SAG1-SAG2-ROP1). The remaining proteins were characterized by slightly lower reactivity: SAG1-SAG2-GRA1 (99.38%), SAG1-SAG2-GRA2 (98.75%), SAG1-SAG2-MAG1 (97.5%), SAG1-SAG2-GRA7 (95%), and SAG1-SAG2-GRA6 (94.38%). The specificity of all IgG ELISA was determined to be 100%. These results confirm our previous findings, where we tested recombinant tetravalent chimeric proteins with serum samples from small ruminants [16]. This work showed that one of the recombinant tetravalent proteins, AMA1-SAG2-GRA1-ROP1, allowed the development of an IgG ELISA test with 96.77% sensitivity and 100% specificity. The literature contains three studies in which single recombinant proteins were examined in both sheep and goats, such as SAG2 [26,27] and GRA7 [28]. However, referencing these works is challenging because, in one of them, the serum samples were not tested with any other test or commercial test based on native antigens [27]. In the remaining two articles, the sensitivity and specificity of the tests in relation to the commercial test performed are not 100%. The IgG ELISA tests based on native antigens were also not performed [26,28]. There are no other studies in the literature conducted by other research teams that compare the usefulness of recombinant proteins for various animal species, in particular for sheep and goats. Moving on to a detailed analysis of the results for individual animal species, we will start with the analysis of the results obtained for sheep. Among the newly produced recombinant trivalent chimeric proteins, nine had a reactivity of 100%, comparable to TLA and the recombinant divalent SAG1-SAG2 protein. For the rest of the recombinant trivalent chimeric proteins, reactivity was recorded at a level between 94.55–98.18%. Our previous work has shown that mixtures M1: GRA1 + ROP1 and M4: GRA1 + SAG2 + ROP1 of recombinant proteins may be an alternative to TLA in the detection of parasite infection in sheep [17]. In our other work, we showed that recombinant trivalent chimeric proteins, such as SAG1_L_-MIC1-MAG1, SAG2-GRA1-ROP1_S_, and SAG2-GRA1-ROP1_L_, can also be an alternative to TLA [15]. In studies by other researchers, single recombinant proteins, such as H4 [29], H11 [29,30], MAG1 [31,32], SAG2 [26,27], and GRA7 [28], and mixtures of two recombinant proteins (H4 + H11 [29] and GRA1 + BAG1 [33]) were tested in IgG ELISAs. The obtained results clearly showed that a single recombinant protein or mixture of two recombinant proteins is not able to correctly recognize all serum samples containing specific anti-*T. gondii* antibodies. Our results once again showed that when constructing recombinant multivalent chimeric proteins, attention should be paid not only to the rational selection of fragments but also to their size. This is very noticeable in the example of recombinant trivalent chimeric proteins, which additionally contained, as a third immunodominant fragment, various sizes of fragments of MAG1, MIC1, and P35 antigens. The use of a shorter immunodominant fragment of antigens may result in the loss of important immunodominant epitopes that are recognized by the B cells of most individuals. When we compare the results obtained for the recombinant divalent chimeric SAG1-SAG2 protein and recombinant trivalent chimeric proteins, we can observe that, in general, the addition of another immunodominant fragment improved immunoreactivity with specific anti-*Toxoplasma* antibodies, which confirms our previous research [2,3,15]. A smaller number of protein preparations were used for research on the diagnostic utility of newly obtained recombinant trivalent chimeric proteins for the detection of *T. gondii* infection in goats. Protein preparations of recombinant trivalent chimeric proteins containing smaller fragments of MAG1, MIC1, and P35 antigens as the third immunodominant fragment were rejected. Recombinant trivalent chimeric proteins containing an immunodominant fragment of the AMA1 and LDH2 antigen in their sequence were also omitted, as the recombinant protein containing a large immunodominant fragment of AMA1 and LDH2 was not characterized by 100% reactivity, while a relatively high cut-off value was determined for protein preparation with the smaller fragment of AMA1 antigen in the IgG ELISA. Of the eleven recombinant trivalent chimeric proteins tested, seven had a reactivity of 100%, comparable to TLA. For the rest of the recombinant trivalent proteins, reactivity was noticed at a level between 90–98%. The sensitivity of the IgG ELISA based on the recombinant divalent chimeric SAG1-SAG2 protein was 98%. In our previous studies, we demonstrated the diagnostic utility of recombinant tetravalent chimeric proteins in the diagnosis of *T. gondii* infection in goats. Two recombinant proteins, AMA1_N_-SAG2-GRA1-ROP1 and AMA1-SAG2-GRA1-ROP1, allowed the development of IgG ELISA tests with 100% specificity and sensitivities of 88.89% and 95.56%, respectively [16]. Other works on the usefulness of recombinant proteins in the diagnosis of *T. gondii* infection in goats concern single recombinant proteins, such as SAG1, SAG2, GRA7, and ROP8 [26,27,28,34,35,36,37], and a mixture of two recombinant proteins, SAG1 + GRA7 [37]. As in the case of sheep, the obtained results showed that a single recombinant protein or mixture of two recombinant proteins is not able to correctly detect infection in all animals.

Considering all the results obtained in this work, we constructed seventeen new trivalent recombinant chimeric proteins. All these recombinant trivalent chimeric proteins reacted with sera collected from experimentally infected laboratory mice. We could observe a typical immune response, which consisted of an increase in the titer of specific anti-*Toxoplasma* IgG antibodies in the tested samples depending on the time after parasite infection. In the case of several protein preparations, the specific IgG antibodies were easily detected from the early stages of parasite invasion. Analyzing the results obtained for two animal species (sheep and goat), it was found that six protein preparations of recombinant trivalent chimeric proteins (SAG1-SAG2-GRA5, SAG1-SAG2-GRA9, SAG1-SAG2-MIC1, SAG1-SAG2-MIC3, SAG1-SAG2-P35, and SAG1-SAG2-ROP1) can be used to develop IgG ELISA tests with 100% sensitivity and specificity, and constitute an alternative to the polyvalent native antigen TLA used in commercial tests. Moreover, the statistical analysis performed showed that the above-mentioned protein preparations are characterized by a strong or very strong correlation coefficient vs. the native antigen TLA. However, it should be remembered that the development of an ideal recombinant protein that could replace the native antigen of the parasite requires research on many animal species. A recombinant chimeric protein useful for diagnosing infection in one species may be completely useless for detecting infection in another [15]. It should be remembered that the immune response between different hosts of the parasite may be completely different, and it often depends on individual characteristics. Genetic diversity between different strains of the parasite, their virulence, ability to invade different hosts, as well as the complexity of the parasite’s life cycle, also affect the possibility of developing new diagnostic tools. Taking into account what serodiagnostic methods are currently used to detect parasite infection in farm animals, the results of this study may constitute a starting point for the development of cheaper tests, e.g., agglutination tests. This would reduce the costs of testing large populations and would finally enable meat labeling. This would make it possible to sell meat marked as free from *T. gondii*, and meat in which the parasite was detected should be subjected to preliminary heat treatment or freezing and then sold as meat safe from *T. gondii* [18,19]. Another solution may be to include information on what type of farm the meat comes from, and whether the farm is run following accepted standards and is subject to strict veterinary control, under which at least a representative group of animals have been examined [38]. Considering that more than 80% of the infected human population became infected as a result of eating raw or undercooked meat, this approach to detecting parasite infection seems to be appropriate. It is estimated that 30–63% of *T. gondii* infections in pregnant women are caused by eating improperly prepared meat [38]. Our results give hope for the development of cheaper diagnostic tools; however, we should also remember to raise public awareness of *T. gondii* infection, along with other parasites as well as fungi, bacteria, and viruses.

## 4. Materials and Methods

### 4.1. T. gondii Single Recombinant Proteins

The twenty-four single recombinant proteins were expressed as previously described (Appendix A). All recombinant proteins contain a cluster of six histidine residues at the N- and C-termini. The proteins were purified using one-step metal affinity chromatography with Ni^2+^ bound to iminodiacetic acid-agarose (Merck, KGaA, Darmstadt, Germany). The purification resulted in electrophoretically homogeneous protein preparations with a purity above 90% (Appendix A). The concentration of purified proteins was determined using Bradford reagent (Merck, KGaA, Darmstadt, Germany) according to the manufacturer’s recommendation.

### 4.2. Ovine and Caprine Serum Samples

Two groups of serum samples derived from 192 sheep and 95 goats were received from the Veterinary Hygiene Station (Gdańsk, Poland). These serum samples have been obtained from epidemiological studies conducted on a farm animal population from the Pomeranian Voivodeship, Poland. All serum samples were analyzed and divided into seropositive and seronegative groups in accordance with the results obtained using the agglutination test (Toxo-Screen DA, bioMérieux, Marcy-l’Étoile, France) and immunofluorescence test, with the use of slides coated with *T. gondii* antigen (Toxo-Spot IF, bioMérieux, Marcy-l’Étoile, France). All serum samples were also seronegative for specific anti-*Neospora caninum* antibodies as determined using a competitive-inhibition enzyme-linked immunosorbent assay (cELISA) (*Neospora caninum* Antibody Test Kit, VMRD, Inc., Pullman, WA, USA). Ovine serum samples were divided into two groups: group I comprised 110 sera from naturally infected sheep (IgG anti-*T. gondii* positive) and group II comprised 82 from seronegative animals (IgG anti-*T. gondii* negative). Caprine serum samples were divided into two groups: group I comprised 50 sera from naturally infected goats and group II comprised 45 from seronegative animals. In the preliminary test, only three seropositive and seronegative serum samples from each group were used. In IgG ELISAs with newly produced recombinant chimeric proteins, the entire pool of serum samples was used.

### 4.3. IgG ELISAs with Single Recombinant Proteins, Recombinant Chimeric Proteins, and TLA

MaxiSorp multi-well plates (Thermo Fisher Scientific, Inc., Waltham, MA, USA) were coated with:(a)single recombinant proteins at a final concentration of 5, 2.5, and 1 μg/mL for each recombinant protein in a coating buffer (0.05 M carbonate buffer, pH 9.6) for the preliminary test;(b)recombinant chimeric proteins at a final concentration of 2.5 μg/mL for each recombinant protein and 1 μg/mL for TLA in a coating buffer (0.05 M carbonate buffer, pH 9.6) for the testing of newly produced recombinant chimeric proteins.

After overnight incubation at 4 °C, the plates were washed three times (PBS, 0.1% Tween 20) and blocked for 1 h at 37 °C in blocking solution (1% bovine serum albumin, 0.5% Tween 20 in PBS). The plates were then washed three times and incubated for 1 h at 37 °C with the animal’s serum diluted 1:100 in blocking solution. Next, the plates were washed three times with washing buffer and, respectively, incubated with anti-sheep or anti-goat IgG peroxidase-labeled conjugates (Jackson ImmunoResearch Europe Ltd., Cambridgeshire, UK) diluted in blocking solution for 1 h at 37 °C. In the preliminary test, two dilutions of secondary antibodies were used, at concentrations of 1:16,000 and 1:32,000, respectively. In studies using newly produced recombinant chimeric proteins, one dilution of secondary antibodies was used at a dilution of 1:32,000. Finally, the plates were washed three times with washing buffer and o-phenylenediamine dihydrochloride chromogenic substrate (Merck, KGaA, Darmstadt, Germany) was added. After 45 min of incubation at 37 °C in darkness, the reaction was stopped by the addition of 2 M sulfuric acid, and the optical density (OD) at 492 nm was measured using a microtiter plate reader (Multiskan FC; Thermo Fisher Scientific, Inc., Waltham, MA, USA).

Each serum sample was examined twice. The results were determined for each sample by calculating the mean OD reading of duplicate wells. A positive result was defined as any value higher than the average OD reading plus 2 standard deviations (cut-off value) obtained with serum samples from control group II, which consisted of seronegative serum samples.

During the titration phase of the tests, the possibility of cross-reactions with BSA/FBS was excluded (the absorbance value read for empty wells, as well as for wells incubated with carbonate buffer, did not exceed 0.100). In order to check the background of the reaction and possible cross-reactions with the purified bacterial lysate, an ELISA test was also performed with all tested sera. For sheep sera, the average absorbance value ± SD for the lysate was 0.212 ± 0.064, while for goat sera, it was 0.222 ± 0.067.

### 4.4. Ethical Statements

All animal experiments have received approval from the Polish Local Ethics Commission for Experiments on Animals in Łódź (Agreement 75/ŁB639/2012), which operates under the Law on the Protection of Animals Used for Scientific or Educational Purposes and conforms to European Directive 2010/63/EU of the European Parliament and of the Council of 22 September 2010 on the protection of animals used for scientific purposes. While preparing the manuscript, the guidelines from ARRIVE were followed.

### 4.5. Parasite

For induction of experimental toxoplasmosis in laboratory mice, a low-virulence *T. gondii* DX strain maintained routinely in vivo was used. The native tachyzoite parasite antigen (TLA) was prepared from the tachyzoites of a highly virulent RH strain of *T. gondii* (PRA-310, ATCC^®^, Manassas, VA, USA) maintained in vitro on human foreskin fibroblasts (Hs27, CRL-1634, ATCC^®^, Manassas, VA, USA).

### 4.6. Animals

The animals used in the experiments were bred in the animal house of the Faculty of Biology and Environmental Protection, University of Lodz, from parental pairs derived from the Charles River Laboratories (Wilmington, MA, USA). The experiments were carried out on BALB/c inbred males aged 8–12 weeks, who were maintained 2–3 per cage in stable conditions with temperatures of 21 °C ± 0.5 to ±5%, 55% humidity, a 12/12 h dark/light cycle, 15–20 air exchanges per hour, and with free access to water and standardized feed.

### 4.7. Mice Infection

Mice were intraperitoneally infected with 5 tissue cysts of the DX strain present in the homogenate of the brain from a mouse chronically infected with *T. gondii*, which was mechanically disrupted by repeated passage through needles with successively smaller diameters. Mouse immune sera (6 per group) were isolated at specified time points post-infection: 2 weeks (acute invasion), 3 weeks (late acute toxoplasmosis), and 6 and 12 weeks (chronic infection). Sera from uninfected animals constituted negative controls.

### 4.8. Preparation of T. gondii Native Lysate Antigen (TLA)

The native whole-cell tachyzoite lysate antigen (TLA) was prepared as described previously [39]. Briefly, maintained in vitro RH strain tachyzoites were lysed through a repeated freeze-thaw technique using liquid nitrogen and a 37 °C water bath. The fraction containing water-soluble *T. gondii* proteins was harvested to serve as a source of native parasite antigens for further evaluation. The concentration of proteins in the preparation was determined with a commercially available Bradford reagent (Merck, KGaA, Darmstadt, Germany) according to the manufacturer’s protocol.

### 4.9. IgG ELISA with Mouse Serum Samples

The usefulness of tested recombinant proteins in the detection of specific anti-*T. gondii* antibodies present in mouse immune sera was evaluated by an indirect ELISA test, as described previously [40]. Briefly, the wells of MaxiSorp (Thermo Fisher Scientific, Inc., Waltham, MA, USA) plates were coated overnight with native or recombinant parasite proteins (1 μg and 0.25 μg in 100 μL /well, respectively). Then, 10% fetal bovine serum (CytoGen, GmbH, Greven, Germany) in PBS (Merck KGaA, Darmstadt, Germany) was used for blocking and both primary and secondary antibody dilution. All tested sera, immune and control, were diluted to 1:100. Secondary goat anti-mouse IgG horseradish peroxidase (HRP)-conjugated antibodies (Jackson ImmunoResearch Europe Ltd., Cambridgeshire, UK) were used at a 1:4000 dilution. The color reaction was developed using H_2_O_2_ (Merck, KGaA, Darmstadt, Germany) and ABTS (Merck, KGaA, Darmstadt, Germany), at the concentration of 1 mg/mL, as a chromogene. After 20 min incubation in darkness, the OD at 405 nm was measured (Multiskan EX ELISA reader, Thermo Fisher Scientific, Inc., Waltham, MA, USA), and mean absorbances of serum samples were calculated. The reactivity of immune serum samples with each antigen was considered positive if the OD exceeded the cut-off value expressed as the mean absorbance of negative control sera + 2 standard deviations.

### 4.10. T. gondii RNA Isolation and cDNA Synthesis

Total RNA from tachyzoites was extracted using Total RNA Mini Plus D (A&A Biotechnology, Gdynia, Poland) according to the manufacturer’s instructions and stored at −80 °C. Single-strand cDNA was synthesized using a TranScriba Kit (A&A Biotechnology, Gdynia, Poland) and stored at −20 °C.

### 4.11. Construction of the Recombinant Plasmids

The DNA encoding appropriate antigen fragments were amplified from the cDNA using a standard PCR amplification protocol with Phusion High-Fidelity DNA Polymerase (Thermo Fisher Scientific, Inc., Waltham, MA, USA). The DNA gene fragments were obtained through PCR using primers as shown in Table 6. In the cases of pET30/SAG1-SAG2, pET30/SAG1-SAG2-AMA1S, pET30/SAG1-SAG2-GRA2, pET30/SAG1-SAG2-LDH2, pET30/SAG1-SAG2-MIC1S, pET30/SAG1-SAG2-MIC3, pET30/SAG1-SAG2-P35S, and pET30/SAG1-SAG2-ROP1, the final PCR products were inserted into the *Bgl*II site of the expression vector pET30 Ek/LIC. In the cases of pET30/SAG1-SAG2-AMA1, pET30/SAG1-SAG2-GRA1, pET30/SAG1-SAG2-GRA5, pET30/SAG1-SAG2-GRA6, pET30/SAG1-SAG2-GRA7, pET30/SAG1-SAG2-GRA9, pET30/SAG1-SAG2-MAG1, pET30/SAG1-SAG2-MAG1S, pET30/SAG1-SAG2-MIC1, and pET30/SAG1-SAG2-P35, the final PCR products were inserted into the *Eco*RV site of the obtained recombinant plasmids pET30/SAG1-SAG2. The In-Fusion HD Cloning Kit (Takara Bio Inc., Kasatsu, Shiga, Japan) based on DNA recombination was used in all cloning reactions.

### 4.12. Production and Purification of Recombinant Chimeric Proteins

*E. coli* strains Rosetta(DE3)pLysS and Rosetta(DE3)pLacI transformed with recombinant plasmids encoding chimeric proteins were grown in Terrific Broth (TB) medium supplemented with 20 µg/mL of kanamycin and 34 µg/mL of chloramphenicol overnight at 30 °C. Next, 400 mL of fresh TB medium, supplemented with the same antibiotics, were inoculated with 8 mL of the overnight culture. The cultures were grown with vigorous shaking at 30 °C until OD_600_ = 0.4. Protein production was then induced with isopropyl-β-D-thiogalactopyranoside (IPTG) at a final concentration of 1 mM, and the bacterial cultures were continued for another 16 h in the same conditions. The bacterial cells were harvested by centrifugation (5000× *g*, 10 min, 4 °C), and the pellets from 100 mL of culture were stored at −20 °C.

All recombinant chimeric proteins were purified using a Ni Sepharose™ 6Fast Flow column (Cytiva, Little Chalfont, UK) following the manufacturer’s protocol. The cell pellets were resuspended in 30 mL of buffer (5 M urea, 20 mM Tris, 0.5 M NaCl, 5 mM imidazole, and 0.1% Triton X-100) and sonicated. After centrifugation (12,000× *g*, 30 min, 4 °C), the protein extracts, obtained by dissolving the inclusion bodies, were purified on columns using buffers with increasing imidazole concentrations. Recombinant chimeric proteins were eluted from the column using a buffer without a denaturing agent containing: 20 mM Tris, 0.5 M NaCl, 0.5 M imidazole, and 0.1% Triton X-100. Recombinant proteins were purified with buffers with a pH of 7.9, except for the SAG1-SAG2-P35 protein, which was purified with buffers with a pH of 9.5.

Molecular weight and PI were calculated using a program available on the website https://web.expasy.org/compute_pi/ (accessed on 12 April 2024).

### 4.13. Statistical Analysis

Statistical analysis was performed using GraphPad Prism 9.1.1 (GraphPad Software, Boston, MA, USA). The t-test was used to assess the statistically significant difference between seropositive serum samples and seronegative serum samples in the test with single recombinant proteins. Receiver operating characteristics (ROC) analysis was performed to obtain the area under the curve (AUC), the sensitivity, and the specificity for the different groups of the compared serum samples. The Pearson product-moment correlation coefficient (*r*) analysis was used to assess the correlation between recombinant chimeric proteins vs. TLA, and recombinant trivalent chimeric proteins vs. the recombinant SAG1-SAG2 protein. Observed correlation coefficients were interpreted as follows: <0.20 = very weak correlation; 0.20–0.49 = weak correlation; 0.50–0.69 = moderate correlation; 0.70–0.89 = strong correlation; and 0.90–1.00 = very strong correlation. Differences with a two-tailed value of *p* < 0.05 were considered statistically significant.

## 5. Conclusions

The development of *T. gondii* recombinant trivalent chimeric proteins as an alternative to TLA in ELISA tests for the detection of specific anti-*T. gondii* IgG antibodies are a significant advancement in the diagnosis of parasite invasion, particularly in small ruminants. The intracellular parasite, *T. gondii*, infects a wide range of warm-blooded animals, including humans. This parasite is widespread among different animal populations, contributes to reproductive losses and malformations in young individuals, and can become a serious economic concern for farmers. Additionally, the consumption of undercooked or raw meat and the consumption of improperly processed milk products derived from farm animals are the main parasite transmission routes in humans. The development of new and cheaper diagnostic tools may allow the development of simple diagnostic algorithms that will soon allow for an effective fight against this dangerous parasitosis.

## Figures and Tables

**Figure 1 ijms-25-04384-f001:**
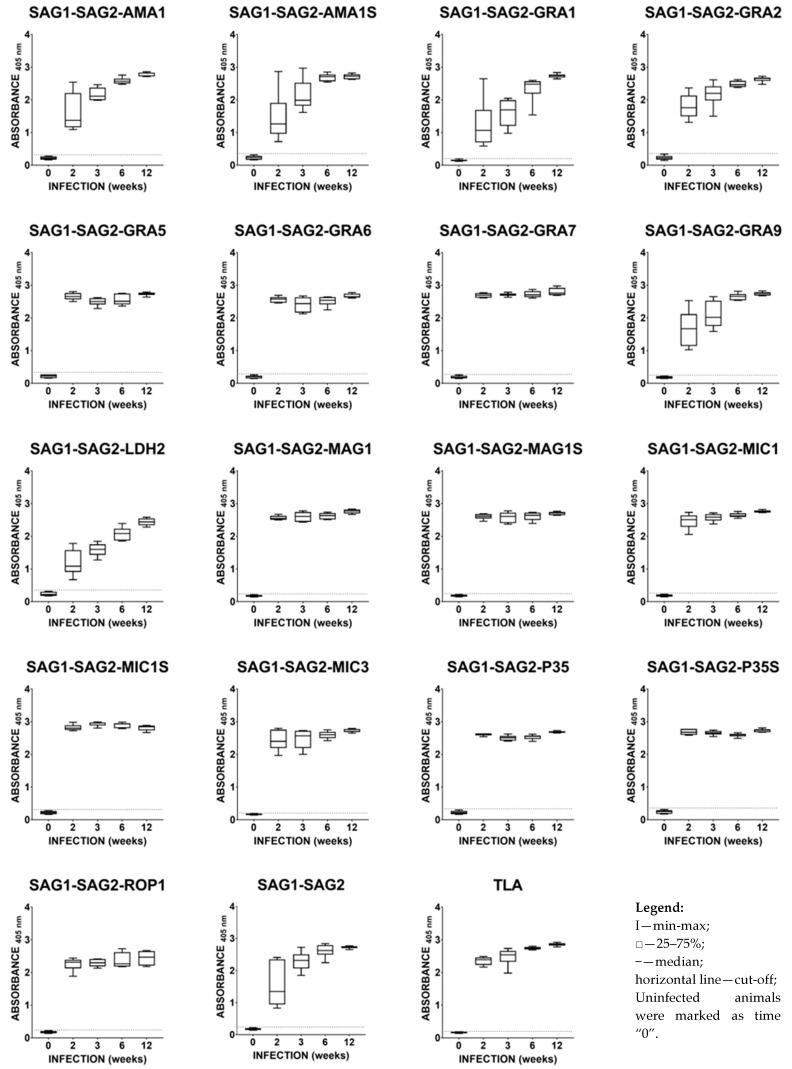
The antibody response of BALB/c mice infected with *T. gondii* DX, tested by IgG ELISA.

**Figure 2 ijms-25-04384-f002:**
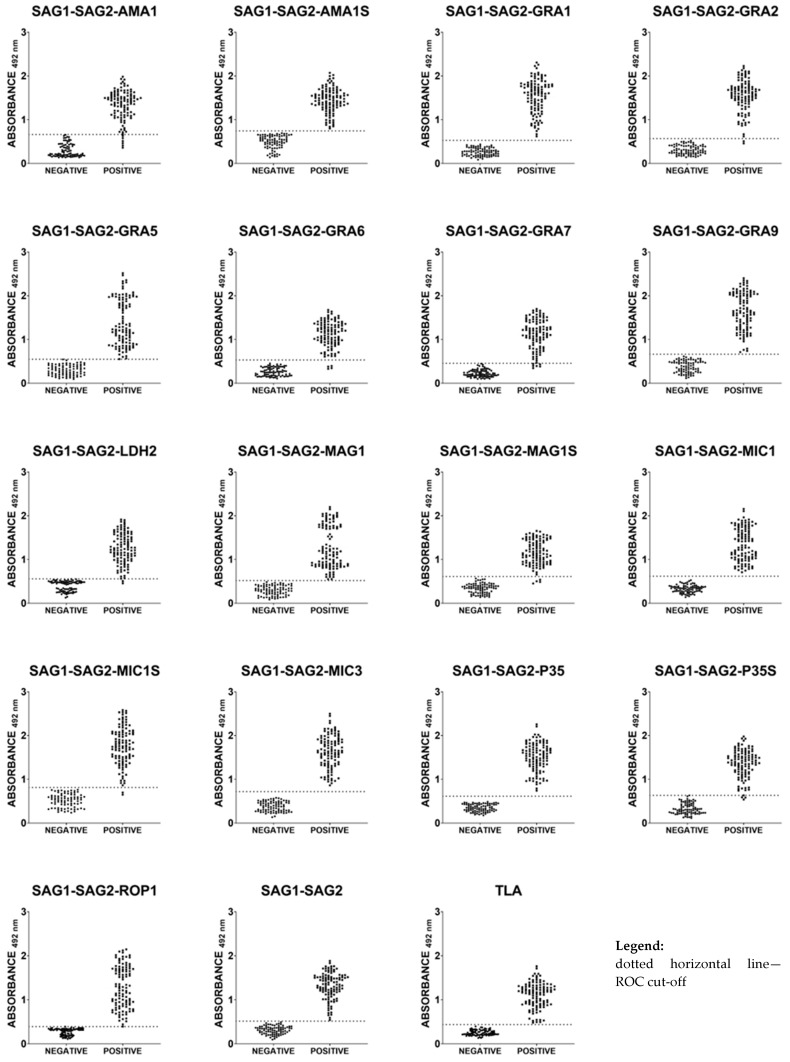
Comparison of immunoreactivity in IgG ELISA using recombinant chimeric proteins and TLA, with 192 ovine serum samples (110 from naturally infected sheep and 82 from seronegative animals).

**Figure 3 ijms-25-04384-f003:**
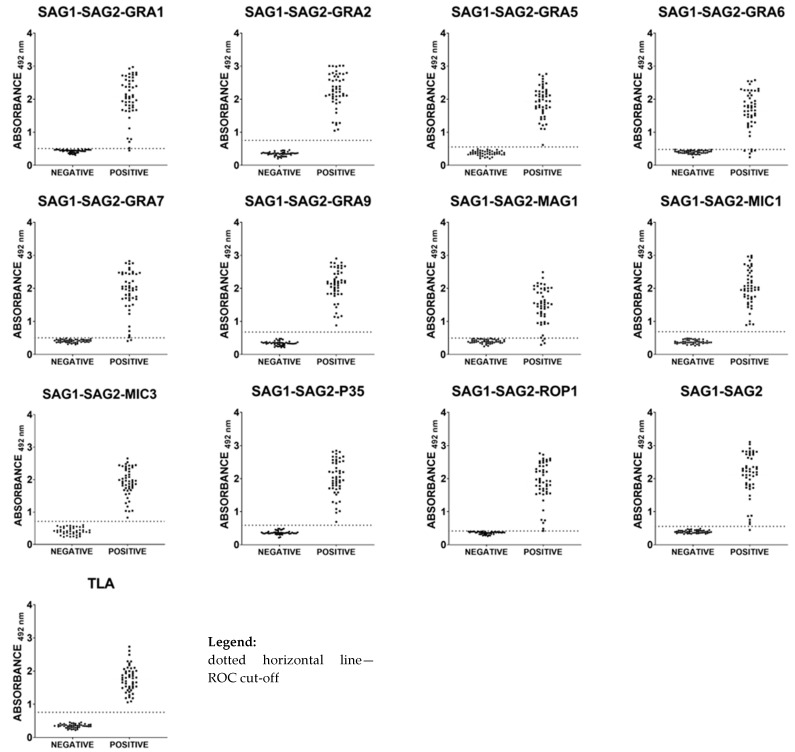
Comparison of immunoreactivity in IgG ELISA using recombinant chimeric proteins and TLA, with 95 caprine serum samples (50 from naturally infected goats and 45 from seronegative animals).

**Table 1 ijms-25-04384-t001:** Absorbance values measured in IgG ELISAs for three seropositive (S+) and three seronegative (S−) ovine and caprine serum samples with twenty-four *T. gondii* single recombinant proteins.

Recombinant Protein	Antibodies Dilution	Animal Species					
		Ovine			Caprine		
		Mean ± SD S+	Mean ± SD S−	*p* Value	Mean ± SD S+	Mean ± SD S−	*p* Value
AMA1	1:16,000	1.360 ± 0.271	0.506 ± 0.047	0.0058 **	2.339 ± 0.414	0.853 ± 0.057	0.0035 **
	1:32,000	0.814 ± 0.175	0.322 ± 0.025	0.0086 **	1.541 ± 0.310	0.522 ± 0.025	0.0047 **
AMA1C	1:16,000	1.124 ± 0.022	0.597 ± 0.153	0.0041 **	1.645 ± 0.232	0.841 ± 0.020	0.0039 **
	1:32,000	0.559 ± 0.062	0.311 ± 0.092	0.0181 *	0.997 ± 0.143	0.545 ± 0.034	0.0060 **
AMA1N	1:16,000	1.087 ± 0.133	0.426 ± 0.064	0.0015 **	2.027 ± 0.388	0.723 ± 0.090	0.0048 **
	1:32,000	0.583 ± 0.100	0.253 ± 0.037	0.0058 **	1.248 ± 0.259	0.410 ± 0.053	0.0053 **
BAG1	1:16,000	1.289 ± 0.314	0.680 ± 0.126	0.0356 *	1.857 ± 0.233	1.354 ± 0.083	0.0245 *
	1:32,000	0.766 ± 0.206	0.395 ± 0.067	0.0411 *	1.088 ± 0.108	0.723 ± 0.021	0.0045 **
BSR4	1:16,000	0.549 ± 0.262	0.223 ± 0.077	0.1076	1.080 ± 0.129	0.766 ± 0.049	0.0168 *
	1:32,000	0.312 ± 0.149	0.151 ± 0.044	0.1468	0.629 ± 0.064	0.464 ± 0.039	0.0186 *
GRA1	1:16,000	1.049 ± 0.144	0.334 ± 0.037	0.0011 **	2.543 ± 0.222	0.599 ± 0.083	0.0001 ***
	1:32,000	0.606 ± 0.078	0.233 ± 0.045	0.0020 **	1.651 ± 0.163	0.350 ± 0.029	0.0002 ***
GRA2ex2	1:16,000	1.411 ± 0.057	0.595 ± 0.138	0.0007 ***	2.456 ± 0.567	0.929 ± 0.246	0.0128 *
	1:32,000	0.880 ± 0.056	0.357 ± 0.061	0.0004 ***	1.558 ± 0.354	0.495 ± 0.133	0.0082 **
GRA4	1:16,000	1.471 ± 0.381	0.850 ± 0.026	0.0482 *	2.667 ± 0.694	1.793 ± 0.221	0.1060
	1:32,000	0.846 ± 0.278	0.505 ± 0.027	0.1020	1.805 ± 0.671	1.042 ± 0.150	0.1265
GRA5	1:16,000	1.550 ± 0.147	0.672 ± 0.018	0.0005 ***	2.291 ± 0.445	0.804 ± 0.063	0.0046 **
	1:32,000	0.905 ± 0.083	0.414 ± 0.013	0.0005 ***	1.394 ± 0.323	0.539 ± 0.093	0.0117 *
GRA6	1:16,000	1.442 ± 0.350	0.498 ± 0.030	0.0096 **	2.218 ± 0.076	0.610 ± 0.061	<0.0001 ****
	1:32,000	0.871 ± 0.224	0.323 ± 0.021	0.0135 *	1.381 ± 0.013	0.432 ± 0.042	<0.0001 ****
GRA7	1:16,000	1.298 ± 0.268	0.382 ± 0.033	0.0042 **	2.370 ± 0.450	1.032 ± 0.034	0.0068 **
	1:32,000	0.845 ± 0.245	0.256 ± 0.021	0.0142 *	1.473 ± 0.308	0.615 ± 0.034	0.0087 **
GRA9	1:16,000	1.255 ± 0.180	0.380 ± 0.039	0.0012 **	2.667 ± 0.694	1.019 ± 0.127	0.0155 *
	1:32,000	0.727 ± 0.114	0.250 ± 0.053	0.0028 **	1.805 ± 0.671	0.560 ± 0.029	0.0325 *
LDH1	1:16 000	1.208 ± 0.674	0.324 ± 0.062	0.0862	1.569 ± 0.311	0.917 ± 0.195	0.0370 *
	1:32,000	0.588 ± 0.309	0.216 ± 0.046	0.1082	1.040 ± 0.345	0.571 ± 0.040	0.0795
LDH2	1:16,000	1.578 ± 0.291	0.565 ± 0.049	0.0040 **	1.847 ± 0.144	0.529 ± 0.061	0.0001 ***
	1:32,000	0.927 ± 0.173	0.353 ± 0.026	0.0047 **	1.168 ± 0.109	0.314 ± 0.020	0.0002 ***
MAG1	1:16,000	1.152 ± 0.304	0.449 ± 0.024	0.0162 *	1.626 ± 0.594	0.644 ± 0.083	0.0470 *
	1:32,000	0.666 ± 0.192	0.277 ± 0.019	0.0251 *	0.954 ± 0.389	0.423 ± 0.048	0.0785
MIC1	1:16,000	1.233 ± 0.216	0.544 ± 0.077	0.0065 **	2.628 ± 0.445	0.885 ± 0.064	0.0026 **
	1:32,000	0.741 ± 0.073	0.335 ± 0.006	0.0007 ***	1.646 ± 0.365	0.475 ± 0.026	0.0052 **
MIC1ex2	1:16,000	1.486 ± 0.042	0.848 ± 0.069	0.0002 ***	2.557 ± 0.418	0.957 ± 0.074	0.0028 **
	1:32,000	0.865 ± 0.040	0.465 ± 0.015	<0.0001 ****	1.565 ± 0.361	0.496 ± 0.084	0.0075 **
MIC1ex3.4	1:16,000	1.105 ± 0.204	0.473 ± 0.138	0.0114 *	2.186 ± 0.497	1.094 ± 0.117	0.0208 *
	1:32,000	0.667 ± 0.137	0.303 ± 0.094	0.0192 *	1.351 ± 0.369	0.640 ± 0.092	0.0317 *
MIC3	1:16,000	1.500 ± 0.357	0.506 ± 0.078	0.0093 **	2.671 ± 0.249	0.999 ± 0.111	0.0004 ***
	1:32,000	0.886 ± 0.187	0.322 ± 0.052	0.0073 **	1.654 ± 0.224	0.640 ± 0.107	0.0021 **
P35	1:16,000	2.410 ± 0.290	1.047 ± 0.054	0.0013 **	1.733 ± 0.080	0.841 ± 0.020	<0.0001 ****
	1:32,000	1.613 ± 0.334	0.514 ± 0.026	0.0047 **	1.062 ± 0.034	0.393 ± 0.020	<0.0001****
ROP1	1:16,000	1.695 ± 0.042	0.541 ± 0.148	0.0002 ***	2.333 ± 0.439	0.800 ± 0.070	0.0039 **
	1:32,000	0.968 ± 0.060	0.338 ± 0.087	0.0005 ***	1.518 ± 0.377	0.491 ± 0.046	0.0094 **
SAG1	1:16,000	1.479 ± 0.430	0.508 ± 0.004	0.0174 *	2.145 ± 0.129	0.865 ± 0.057	<0.0001 ****
	1:32,000	0.873 ± 0.268	0.299 ± 0.007	0.0207 *	1.189 ± 0.117	0.534 ± 0.046	0.0008 ***
SAG2	1:16,000	1.885 ± 0.324	0.634 ± 0.028	0.0026 **	2.625 ± 0.330	0.718 ± 0.046	0.0006 ***
	1:32,000	1.118 ± 0.183	0.368 ± 0.015	0.0021 **	1.639 ± 0.294	0.518 ± 0.061	0.0030 **
SAG4	1:16,000	1.454 ± 0.527	0.700 ± 0.122	0.0730	1.625 ± 0.171	1.000 ± 0.025	0.0033 **
	1:32,000	0.908 ± 0.369	0.444 ± 0.042	0.0961	0.974 ± 0.109	0.586 ± 0.017	0.0037 **

* *p* ≤ 0.05, ** *p* ≤ 0.01, *** *p* ≤ 0.001, **** *p* ≤ 0.0001.

**Table 2 ijms-25-04384-t002:** Characteristics of recombinant chimeric proteins.

Recombinant Chimeric Protein	Amino Acid Residues	Additional Immunodominant Fragment	Amino Acid Residues	Number of Amino Acid Residues	Mw [kDa]	pI
SAG1-SAG2-AMA1	SAG1 amino acid residues from 49-310 SAG2 amino acid residues from 30-170	AMA1	from 67-568	983	106.08	5.78
SAG1-SAG2-AMA1S		AMA1S	from 67-483	898	96.76	5.97
SAG1-SAG2-GRA1		GRA1	from 24-190	648	68.05	5.00
SAG1-SAG2-GRA2		GRA2	from 51-185	616	64.84	6.60
SAG1-SAG2-GRA5		GRA5	from 26-120	576	60.53	5.93
SAG1-SAG2-GRA6		GRA6	from 30-228	681	70.92	5.75
SAG1-SAG2-GRA7		GRA7	from 27-236	691	73.38	5.61
SAG1-SAG2-GRA9		GRA9	from 21-318	779	83.43	5.69
SAG1-SAG2-LDH2		LDH2	from 2-326	806	85.33	6.26
SAG1-SAG2-MAG1		MAG1	from 30-452	904	96.34	5.24
SAG1-SAG2-MAG1S		MAG1S	from 30-222	674	70.82	5.09
SAG1-SAG2-MIC1		MIC1	from 25-456	913	96.30	5.71
SAG1-SAG2-MIC1S		MIC1ex2	from 25-182	639	67.75	6.75
SAG1-SAG2-MIC3		MIC3	from 67-359	774	81.09	6.09
SAG1-SAG2-P35		P35	from 26-377	834	88.13	8.98
SAG1-SAG2-P35S		P35S	from 26-170	626	65.28	6.28
SAG1-SAG2-ROP1		ROP1	from 85-396	793	83.69	6.07
SAG1-SAG2		-	-	481	50.15	6.15

**Table 3 ijms-25-04384-t003:** ROC analysis of the results obtained in the IgG ELISA using sheep sera.

Antigen	Calculated Cut-Off	ROC Cut-Off	Sensitivity [%] (95% CI)	Specificity [%] (95% CI)	Area under the Curve (95% CI)	*p* Value
SAG1-SAG2-AMA1	0.5962	0.6630	94.55 (88.61–97.48)	100 (95.52–100)	0.9909 (0.9816–1.000)	<0.0001
SAG1-SAG2-AMA1S	0.7912	0.7450	100 (96.63–100)	100 (95.52–100)	1.000 (1.000–1.000)	<0.0001
SAG1-SAG2-GRA1	0.4376	0.5278	100 (96.63–100)	100 (95.52–100)	1.000 (1.000–1.000)	<0.0001
SAG1-SAG2-GRA2	0.5133	0.5690	98.18 (93.61–99.68)	100 (95.52–100)	0.9990 (0.9970–1.000)	<0.0001
SAG1-SAG2-GRA5	0.5271	0.5485	100 (96.63–100)	100 (95.52–100)	1.000 (1.000–1.000)	<0.0001
SAG1-SAG2-GRA6	0.4525	0.5320	96.36 (91.02–98.58)	100 (95.52–100)	0.9928 (0.9847–1.000)	<0.0001
SAG1-SAG2-GRA7	0.3914	0.4568	94.55 (88.61–97.48)	100 (95.52–100)	0.9978 (0.9949–1.000)	<0.0001
SAG1-SAG2-GRA9	0.6491	0.6643	100 (96.63–100)	100 (95.52–100)	1.000 (1.000–1.000)	<0.0001
SAG1-SAG2-LDH2	0.6246	0.5573	98.18 (93.61–99.68)	100 (95.52–100)	0.9990 (0.9970–1.000)	<0.0001
SAG1-SAG2-MAG1	0.5095	0.5178	100 (96.63–100)	100 (95.52–100)	1.000 (1.000–1.000)	<0.0001
SAG1-SAG2-MAG1S	0.5563	0.6113	96.36 (91.02–98.58)	100 (95.52–100)	0.9971 (0.9934–1.000)	<0.0001
SAG1-SAG2-MIC1	0.5001	0.6185	100 (96.63–100)	100 (95.52–100)	1.000 (1.000–1.000)	<0.0001
SAG1-SAG2-MIC1S	0.8130	0.8130	98.18 (93.61–99.68)	100 (95.52–100)	0.9968 (0.9919–1.000)	<0.0001
SAG1-SAG2-MIC3	0.6005	0.7188	100 (96.63–100)	100 (95.52–100)	1.000 (1.000–1.000)	<0.0001
SAG1-SAG2-P35	0.5134	0.6135	100 (96.63–100)	100 (95.52–100)	1.000 (1.000–1.000)	<0.0001
SAG1-SAG2-P35S	0.6213	0.6323	97.27 (92.29–99.26)	100 (95.52–100)	0.9991 (0.9976–1.000)	<0.0001
SAG1-SAG2-ROP1	0.4276	0.3915	100 (96.63–100)	100 (95.52–100)	1.000 (1.000–1.000)	<0.0001
SAG1-SAG2	0.4910	0.5168	100 (96.63–100)	100 (95.52–100)	1.000 (1.000–1.000)	<0.0001
TLA	0.3737	0.4378	100 (96.63–100)	100 (95.52–100)	1.000 (1.000–1.000)	<0.0001

**Table 4 ijms-25-04384-t004:** ROC analysis of the results obtained in the IgG ELISA using goat sera.

Antigen	Calculated Cut-Off	ROC Cut-Off	Sensitivity [%] (95% CI)	Specificity [%] (95% CI)	Area under the Curve (95% CI)	*p* Value
SAG1-SAG2-GRA1	0.5249	0.5095	98 (89.50–99.90)	100 (92.13–100)	0.9924 (0.9773–1.000)	<0.0001
SAG1-SAG2-GRA2	0.4680	0.7560	100 (92.87–100)	100 (92.13–100)	1.000 (1.000–1.000)	<0.0001
SAG1-SAG2-GRA5	0.5077	0.5558	100 (92.87–100)	100 (92.13–100)	1.000 (1.000–1.000)	<0.0001
SAG1-SAG2-GRA6	0.5021	0.4810	90 (78.64–95.56)	100 (92.13–100)	0.9467 (0.8938–0.9995)	<0.0001
SAG1-SAG2-GRA7	0.4981	0.5053	96 (86.54–98.36)	100 (92.13–100)	0.9829 (0.9581–1.000)	<0.0001
SAG1-SAG2-GRA9	0.4721	0.6790	100 (92.87–100)	100 (92.13–100)	1.000 (1.000–1.000)	<0.0001
SAG1-SAG2-MAG1	0.5132	0.4978	92 (81.16–96.85)	100 (92.13–100)	0.9556 (0.9050–1.000)	<0.0001
SAG1-SAG2-MIC1	0.4992	0.6893	100 (92.87–100)	100 (92.13–100)	1.000 (1.000–1.000)	<0.0001
SAG1-SAG2-MIC3	0.6300	0.7129	100 (92.87–100)	100 (92.13–100)	1.000 (1.000–1.1000)	<0.0001
SAG1-SAG2-P35	0.4795	0.5898	100 (92.87–100)	100 (92.13–100)	1.000 (1.000–1.000)	<0.0001
SAG1-SAG2-ROP1	0.4357	0.4158	100 (92.87–100)	100 (92.13–100)	1.000 (1.000–1.000)	<0.0001
SAG1-SAG2	0.4799	0.5555	98 (89.50–99.90)	100 (92.13–100)	0.9973 (0.9915–1.000)	<0.0001
TLA	0.4652	0.7580	100 (92.87–100)	100 (92.13–100)	1.000 (1.000–1.000)	<0.0001

**Table 5 ijms-25-04384-t005:** Pearson product-moment correlation coefficient (*r*) between results obtained for recombinant chimeric proteins vs. TLA results, and recombinant trivalent chimeric proteins vs. recombinant SAG1-SAG2 divalent chimeric proteins.

Recombinant Chimeric Protein		Sheep Serum Samples			Goat Serum Samples		
		*r* (95% CI)	*r^2^*	*p* Value	*r* (95% CI)	*r^2^*	*p* Value
SAG1-SAG2-GRA1	vs. TLA vs. SAG1-SAG2	0.9301 (0.9081–0.9470) 0.9254 (0.9019–0.9433)	0.8651 0.8563	<0.0001 <0.0001	0.9634 (0.9455–0.9756) 0.9867 (0.9081–0.9912)	0.9282 0.9737	<0.0001 <0.0001
SAG1-SAG2-GRA2	vs. TLA vs. SAG1-SAG2	0.9254 (0.9020–0.9434) 0.9584 (0.9451–0.9686)	0.8564 0.9186	<0.0001 <0.0001	0.9635 (0.9456–0.9756) 0.9865 (0.9798–0.9910)	0.9284 0.9733	<0.0001 <0.0001
SAG1-SAG2-GRA5	vs. TLA vs. SAG1-SAG2	0.7883 (0.7279–0.8365) 0.7851 (0.7240–0.8340)	0.6214 0.6164	<0.0001 <0.0001	0.9659 (0.9492–0.9772) 0.9762 (0.9643–0.9841)	0.9330 0.9529	<0.0001 <0.0001
SAG1-SAG2-GRA6	vs. TLA vs. SAG1-SAG2	0.9372 (0.9174–0.9524) 0.9328 (0.9116–0.9491)	0.8784 0.8702	<0.0001 <0.0001	0.9119 (0.8703–0.9406) 0.9777 (0.9667–0.9851)	0.8316 0.9559	<0.0001 <0.0001
SAG1-SAG2-GRA7	vs. TLA vs. SAG1-SAG2	0.8883 (0.8542–0.9149) 0.9287 (0.9063–0.9459)	0.7892 0.8625	<0.0001 <0.0001	0.9383 (0.9086–0.9586) 0.9900 (0.9850–0.9933)	0.8804 0.9801	<0.0001 <0.0001
SAG1-SAG2-GRA9	vs. TLA vs. SAG1-SAG2	0.8750 (0.8371–0.9045) 0.8763 (0.8388–0.9056)	0.7656 0.7680	<0.0001 <0.0001	0.9680 (0.9523–0.9786) 0.9857 (0.9786–0.9905)	0.9371 0.9717	<0.0001 <0.0001
SAG1-SAG2-MAG1	vs. TLA vs. SAG1-SAG2	0.8091 (0.7539–0.8530) 0.7708 (0.7063–0.8226)	0.6546 0.5941	<0.0001 <0.0001	0.9031 (0.8577–0.9345) 0.9803 (0.9704–0.9868)	0.8156 0.9609	<0.0001 <0.0001
SAG1-SAG2-MIC1	vs. TLA vs. SAG1-SAG2	0.8774 (0.8402–0.9064) 0.8578 (0.8153–0.8912)	0.7698 0.7359	<0.0001 <0.0001	0.9618 (0.9430–0.9744) 0.9827 (0.9741–0.9885)	0.9250 0.9657	<0.0001 <0.0001
SAG1-SAG2-MIC3	vs. TLA vs. SAG1-SAG2	0.9413 (0.9227–0.9555) 0.9592 (0.9461–0.9692)	0.8860 0.9201	<0.0001 <0.0001	0.9565 (0.9352–0.9709) 0.9677 (0.9759–0.9892)	0.9149 0.9677	<0.0001 <0.0001
SAG1-SAG2-P35	vs. TLA vs. SAG1-SAG2	0.9487 (0.9323–0.9612) 0.9609 (0.9483–0.9705)	0.9000 0.9233	<0.0001 <0.0001	0.9667 (0.9503–0.9777) 0.9861 (0.9792–0.9908)	0.9344 0.9724	<0.0001 <0.0001
SAG1-SAG2-ROP1	vs. TLA vs. SAG1-SAG2	0.8082 (0.7528–0.8523) 0.8005 (0.7432–0.8462)	0.6532 0.6409	<0.0001 <0.0001	0.9657 (0.9489–0.9771) 0.9871 (0.9807–0.9914)	0.9326 0.9744	<0.0001 <0.0001
SAG1-SAG2	vs. TLA	0.9260 (0.9027–0.9438)	0.8574	<0.0001	0.9388 (0.9093–0.9589)	0.8813	<0.0001

**Table 6 ijms-25-04384-t006:** Oligonucleotide primers used for the amplification of gene fragments.

Recombinant Chimeric Protein	Gene Fragment	Primer Sequence	Corresponding to Protein Residues
SAG1-SAG2-AMA1	*ama1*	SS-AMA1For 5′-CTCAACCATGGCGATCACGTCGGGGAATCCCTTTCA-3′ SS-AMA1Rev 5′-GAATTCGGATCCGATTCCCCCTCGACCATAACATGTG-3′	67-568
SAG1-SAG2-AMA1S	*sag1*	SS-SAG1For 5′-TGGACAGCCCAGATCCGGATCCCCCTCTTGTTGC-3′ SS-SAG1Rev 5′-TGGGCGCTGGCGTCTCAGCCGATTTTGCTGAC-3′	49-310
	*sag2*	SS-SAG2For 5′-GTCAGCAAAATCGGCTGAGACGCCAGCGCCCA-3′ SS-S2/AMA1For 5′- AGGGATTCCCCGACGTCGTGAGAGACACAGGG-3′	30-170
	*ama1*	SSA-AMA1For 5′-CCCTGTGTCTCTCACGACGTCGGGGAATCCCT-3′ SSA-AMA1Rev 5′-ATCGGTACCCAGATCAGTGTTAGAGCCACATTCATTTTGTTCG-3′	67-483
SAG1-SAG2-GRA1	*gra1*	SS-GRA1For 5′-CTCAACCATGGCGATCGCTGCCGAAGGCG-3′ SS-GRA1Rev 5′-GAATTCGGATCCGATTCTCTCTCTCCTGTTAGGAACCCAAT-3′	24-190
SAG1-SAG2-GRA2	*sag1*	SS-SAG1For 5′-TGGACAGCCCAGATCCGGATCCCCCTCTTGTTGC-3′ SS-SAG1Rev 5′-TGGGCGCTGGCGTCTCAGCCGATTTTGCTGAC-3′	49-310
	*sag2*	SS-SAG2For 5′-GTCAGCAAAATCGGCTGAGACGCCAGCGCCCA-3′ SS-S2/GRA2Rev 5′-GGTGTATGTTCACCTTTTCCCGTGAGAGACACAGGGTC-3′	30-170
	*gra2*	SSG-GRA2For 5′-GACCCTGTGTCTCTCACGGGAAAAGGTGAACATACACC-3′ SSG-GRA2Rev 5′-ATCGGTACCCAGATCCTGCGAAAAGTCTGGGACGG-3′	51-185
SAG1-SAG2-GRA5	*gra5*	SS-GRA5For 5′-CTCAACCATGGCGATCGGTTCAACGCGTGACG-3′ SS-GRA5Rev 5′-GAATTCGGATCCGATTCTTCCTCGGCAACTTCTTCCT-3′	26-120
SAG1-SAG2-GRA6	*gra6*	SS-GRA6For 5′-CTCAACCATGGCGATCATGGGTGTACTCGTCAATTCGTTG-3′ SS-GRA6Rev 5′-GAATTCGGATCCGATTCAAACACATTCACACGTTCCGG-3′	30-228
SAG1-SAG2-GRA7	*gra7*	SS-GRA7For 5′-CTCAACCATGGCGATGGCCACCGCGTCAGAT-3′ SS-GRA7Rev 5′-GAATTCGGATCCGATTGGCGGGCATCCTCCC-3′	27-236
SAG1-SAG2-GRA9	*gra9*	SS-GRA9For 5′-CTCAACCATGGCGATACTCGACCTTTTCCTCGGTGAA-3′ SS-GRA9Rev 5′-GAATTCGGATCCGATAGTCCTCGGTCTTCCTGCG-3′	21-318
SAG1-SAG2-LDH2	*sag1*	SS-SAG1For 5′-TGGACAGCCCAGATCCGGATCCCCCTCTTGTTGC-3′ SS-SAG1Rev 5′-TGGGCGCTGGCGTCTCAGCCGATTTTGCTGAC-3′	49-310
	*sag2*	SS-SAG2For 5′-GTCAGCAAAATCGGCTGAGACGCCAGCGCCCA-3′ SSL-S2/LDH2Rev 5′-TGCTAACGGTACCCGTCGTGAGAGACACAGGG-3′	30-170
	*ldh2*	SSL-LDH2For 5′-CCCTGTGTCTCTCACGACGGGTACCGTTAGCA-3′ SSL-LDH2Rev 5′-ATCGGTACCCAGATCACCCAGCGCCGCT-3′	2-326
SAG1-SAG2-MAG1	*mag1*	SS-MAG1For 5′-CTCAACCATGGCGATGAGCCAAAGGGTGCCAGAG-3′ SS-MAG1Rev 5′-GAATTCGGATCCGATGCTGCCTGTTCCGCTAAGAT-3′	30-452
SAG1-SAG2-MAG1S	*mag1*	SS-MAG1For 5′-CTCAACCATGGCGATGAGCCAAAGGGTGCCAGAG-3′ SS-MAG1SRev 5′-GAATTCGGATCCGATTTCTTGATGGCTTCCAACTGCT-3′	30-222
SAG1-SAG2-MIC1	*mic1*	SS-MIC1For 5′-CTCAACCATGGCGATAGCGTCGCATTCTCATTCGC-3′ SS-MIC1Rev 5′-GAATTCGGATCCGATGCAGAGACGGCCGTAGG-3′	25-456
SAG1-SAG2-MIC1S	*sag1*	SS-SAG1For 5′-TGGACAGCCCAGATCCGGATCCCCCTCTTGTTGC-3′ SS-SAG1Rev 5′-TGGGCGCTGGCGTCTCAGCCGATTTTGCTGAC-3′	49-310
	*sag2*	SS-SAG2For 5′-GTCAGCAAAATCGGCTGAGACGCCAGCGCCCA-3′ SSM-S2/MIC1Rev 5′-CGAATGAGAATGCGACGCCGTGAGAGACACAGGGT-3′	30-170
	*mic1*	SSM-MIC1For 5′-ACCCTGTGTCTCTCACGGCGTCGCATTCTCATTCG-3′ SSM-MIC1Rev 5′-ATCGGTACCCAGATCCTTCTCGTAACACCTCCACGCA-3′	25-182
SAG1-SAG2-MIC3	*sag1*	SS-SAG1For 5′-TGGACAGCCCAGATCCGGATCCCCCTCTTGTTGC-3′ SS-SAG1Rev 5′-TGGGCGCTGGCGTCTCAGCCGATTTTGCTGAC-3′	49-310
	*sag2*	SS-SAG2For 5′-GTCAGCAAAATCGGCTGAGACGCCAGCGCCCA-3′ SSM-S2/MIC3Rev 5′-CCTGCTTGCTGGGGGACGTGAGAGACACAGG-3′	30-170
	*mic3*	SSM-MIC3For 5′-CCTGTGTCTCTCACGTCCCCCAGCAAGCAGG-3′ SSM-MIC3Rev 5′-ATCGGTACCCAGATCCTGCTTAATTTTCTCACACGTCACGG-3′	67-359
SAG1-SAG2-P35	*p35*	SS-P35For 5′-CTCAACCATGGCGATCGGTCCTTTGAGTTATCATCCAAGC-3′ SS-P35Rev 5′-GAATTCGGATCCGATTTCTGCGTCGTTACGGTGAATCT-3′	26-377
SAG1-SAG2-P35S	*sag1*	SS-SAG1For 5′-TGGACAGCCCAGATCCGGATCCCCCTCTTGTTGC-3′ SS-SAG1Rev 5′-TGGGCGCTGGCGTCTCAGCCGATTTTGCTGAC-3′	49-310
	*sag2*	SS-SAG2For 5′-GTCAGCAAAATCGGCTGAGACGCCAGCGCCCA-3′ SSP-S2/P35Rev 5′-GATGATAACTCAAAGGACCCGTGAGAGACACAGGGTC-3′	30-170
	*p35*	SSP-P35For 5′-GACCCTGTGTCTCTCACGGGTCCTTTGAGTTATCATC-3′ SSP-P35Rev 5′-ATCGGTACCCAGATCAGCAGCTGTCGTGGTTGT-3′	26-170
SAG1-SAG2-ROP1	*sag1*	SS-SAG1For 5′-TGGACAGCCCAGATCCGGATCCCCCTCTTGTTGC-3′ SS-SAG1Rev 5′-TGGGCGCTGGCGTCTCAGCCGATTTTGCTGAC-3′	49-310
	*sag2*	SS-SAG2For 5′-GTCAGCAAAATCGGCTGAGACGCCAGCGCCCA-3′ SSR-S2/ROP1Rev 5′-CGGGCCTCTGACAGGCGTGAGAGACACAGG-3′	30-170
	*rop1*	SSR-ROP1For 5′-CCTGTGTCTCTCACGCCTGTCAGAGGCCCG-3′ SSR-ROP1Rev 5′-ATCGGTACCCAGATCTTGCGATCCATCATCCTGCTCTC-3′	85-396
SAG1-SAG2	*sag1*	SS-SAG1For 5′-TGGACAGCCCAGATCCGGATCCCCCTCTTGTTGC-3′ SS-SAG1Rev 5′-TGGGCGCTGGCGTCTCAGCCGATTTTGCTGAC-3′	49-310
	*sag2*	SS-SAG2For 5′-GTCAGCAAAATCGGCTGAGACGCCAGCGCCCA-3′ SS-SAG2Rev 5′-ATCGGTACCCAGATCCGTGAGAGACACAGGGTCAAAC-3′	30-170

## Data Availability

All data generated and analyzed during this study that supports the findings are included in this published article/Appendix A. Further inquiries can be directed to the corresponding author.

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
