# Peer review of "The Development of Toxoplasma gondii Recombinant Trivalent Chimeric Proteins as an Alternative to Toxoplasma Lysate Antigen (TLA) in Enzyme-Linked Immunosorbent Assay (ELISA) for the Detection of Immunoglobulin G (IgG) in Small Ruminants"

_ijms, 2024, doi:10.3390/ijms25084384_

Round 1

Reviewer 1 Report

Comments and Suggestions for Authors

The manuscript “ The development of Toxoplasma gondii recombinant trivalent  chimeric proteins as an alternative to Toxoplasma lysate antigen  (TLA) in enzyme-linked immunosorbent assay (ELISA) for the detection of immunoglobulin G (IgG) in the diagnosis of parasite invasion in small ruminants” by Ferra et al., evaluated recombinant trivalent chimeric proteins, containing the fragments of T. gondii SAG1 and SAG2 antigens, together with an additional immunodominant fragment of other parasite antigens, as a potential alternative to the whole-cell tachyzoite lysate (TLA) used in the detection of infection in small ruminants by IgG ELISA. Six trivalent chimeric proteins (SAG1-SAG2-GRA5, SAG1-SAG2-GRA9, SAG1-SAG2-MIC1, SAG1- SAG2-MIC3, SAG1-SAG2-P35, and SAG1-SAG2-ROP1) were selected and their sensitivity was comparable to the sensitivity of TLA-based IgG ELISA (100%).

 In the manuscript is indicated that TLA shows 100% specificity and 100% sensitivity. The manuscript is of interest in its field, However, it is very long. Both the introduction and discussion sections of the manuscripts are too long and should be reduced to focus on what is important for the study background in recombinant proteins and serological studies in small ruminants. Which is the level of sensitivity the authors needed for the recombinant antigens to be used in the diagnoses in field conditions?

Main comments:

Title. Eliminate “parasite invasion”. Serological analysis of IgG is only indication of contact with the parasite,

Introduction: Lines 43-79: Authors need to reduce significantly the general introduction of T. gondii in humans. Focus on previous studies in small ruminants.

Line 87-89. The authors mention that one limitation of the TLA is that fails to distinguish between the acute and chronic phases of toxoplasmosis, which is of paramount importance, especially for pregnant women. How will the recombinant proteins used in the present study for IgG detection allow to discriminate acute and chronic phases of toxoplasmosis? Please explain.

Lines 92-100 could be eliminated.

Lines: 112-113. Many recombinant proteins have been developed by the same authors “that can potentially be used as an alternative to TLA in the serodiagnosis of T. gondii invasion in humans [16] and animals [17]” Are there any commercial ELISA based on those recombinant proteins? And if so, in which species? What will be the next steps for such a commercial ELISA based on recombinant proteins?

Lines 119-138 reduce significantly.

Line 116-117. The author developed recombinant chimeric proteins composed of various immunodominant fragments of different parasite antigens, were those in other species, not for small ruminants? Please clarify.

As indicated above, the authors could briefly indicate their previous studies in small ruminants and the reasons the present study is needed.

Discussion: Reduce greatly to focus on the main topic of serological diagnosis in small ruminant.

 Lines 354-359: “indirect serological tests such as ELISA have not been widely used in the diagnosis of parasite infection in animals. This is probably due to the need for specialized equipment such as a microplate washer and a microplate reader, as well as the use of  species-specific secondary antibodies. How are these limitations different for ELISA tests using recombinant proteins?

Lines 371-421. Eliminate or reduce significantly citing other reviews/studies. Not the focus of the present study.

The study indicated in Lines 435-438 are of interest since it was also performed by the authors in a previous study in small animals. The advantages and disadvantages of tetravalent proteins versus the trivalent antigens used in the present study should be explained.

There are a lot of references in the manuscript, including also many previous studies for the same authors. Please reduce. 

Reviewer 2 Report

Comments and Suggestions for Authors

The study titled “The development of Toxoplasma gondii recombinant trivalent 2 chimeric proteins as an alternative to Toxoplasma lysate antigen 3 (TLA) in enzyme-linked immunosorbent assay (ELISA) for the 4 detection of immunoglobulin G (IgG) in the diagnosis of para-5 site invasion in small ruminants” by Ferra et al is a novel idea and is well designed. The authors have performed significant amount of work and shown that we can use chimeric proteins instead of TLA for better diagnosis. However, there are several drawbacks in the study with missing controls, poor gel pictures, insufficient statistical analysis and protein characterization. I recommend the authors to address the comments below to improve the manuscript and the study.

1.      The results showed in table 1 and 2 do not have positive and negative control. This is mandatory for an ELISA study. The authors should also the results for binding of sera to bacterial lysate without any protein being expressed. This should serve as the second negative control to prove that the sera dose not bind to any E. coli proteins.

2.      Throughout the study, the authors used BSA of FBS as blocking agents, both of which are derived from bovine. Bovines being closely related to ovine and caprine, contains similar antibody repertoire. There is a chance of cross-reactivity of BSA protein and antibodies present in FBS. The authors should perform control studies to prove that there is no cross-reactivity.

3.      Table 1 and 2: The authors should merge table 1 and 2. Instead of providing the OD value for each sera sample, the authors should give the mean of three sera reads and their SEM/SD. Showing the OD values for every read does not provide any additional meaning.

4.      The authors should show statistically significant difference between seropositive serum samples and seronegative serum samples for values in table 1 and 2. I would recommend ANOVA or t-test.

5.      Move table 3 to supplementary data. This is trivial information and need not be accommodated in the results.

6.      Table 4: How were the molecular weight and PI determined so accurately without performing or validating with experiments. If this was determined by servers, it should be mentioned as predicted molecular weight and PI. The server used should also be added to methodology.

7.      Figure 1: The SDS PAGE image looks pixelated and modified. The bands are not clearly visible. The authors should provide a figure with clear bands visible.

8.      The authors mention that the proteins were metal-affinity purified. The authors should therefore show a band where the purified proteins are shown with purity. The current figure 1 is bacterial lysates with poor image resolution which is not acceptable.

9.      The authors should refer to standard journal articles and reframe the figure caption for figure 1. Only the construct name should be give ie.e sag1-sag-2-xxxx. The information provided now is not required and is space filler.

10.   The authors claim that they used immunodominant fragments. How did the authors confirm that it is immunodominant? Just because the IgG response is higher than SAG1-SAG2, the other constructs cannot be called immunodominant. The presence of antibodies towards another protein will naturally increase the repertoire to target the new domain also. There are no immunological studies performed to show this. Therefore, the authors should reconsider the statement throughout.

11.   Figure 2, table 5, figure 3, table 6, figure 4, table 7: the dilution of the sera use is not mentioned. There are no controls shown. Bacterial lysate should be without any protein expression should be a control, blocking solution another control, bacterial lysate with empty plasmid should be another control. Without controls, the data is obsolete.

12.   Figure 2 and other figures: the y-axis is labelled OD405 nm whereas the ELISA results were recorded at 495 as mentioned in methodology. Why is there a discrepancy with this?

13.   Table 8 should be shifted to supplementary data.

14.   In the methodology, were the proteins dialysed? Since there is urea used, the authors should have dialyzed the protein.

15.   The authors have made a chimeric protein construct in this study. Did the authors use any linkers to stitch the proteins? Sine the abs produced in the animals are towards natural proteins which are structured, here the authors have not ensured any steps to protect or validate the structure. This needs to be justified.

Comments on the Quality of English Language

Minor proof reading required. 

Round 2

Reviewer 1 Report

Comments and Suggestions for Authors

The authors made all the necessary changes, and the manuscript can now be considered acceptable for publication.

Reviewer 2 Report

Comments and Suggestions for Authors

The authors have addressed most of my comments satisfactorily.